# Patterning precision under non-linear morphogen decay and molecular noise

**Jan Andreas Adelmann[1,2], Roman Vetter[1,2], Dagmar Iber[1,2]***

[1]Department of Biosystems Science and Engineering, ETH Zurich, Basel, Switzerland;
[2]Swiss Institute of Bioinformatics, Basel, Switzerland

**Abstract** Morphogen gradients can instruct cells about their position in a patterned tissue. Non-linear morphogen decay has been suggested to increase gradient precision by reducing the sensitivity to variability in the morphogen source. Here, we use cell-based simulations to quantitatively compare the positional error of gradients for linear and non-linear morphogen decay. While we confirm that non-linear decay reduces the positional error close to the source, the reduction is very small for physiological noise levels. Far from the source, the positional error is much larger for non-linear decay in tissues that pose a flux barrier to the morphogen at the boundary. In light of this new data, a physiological role of morphogen decay dynamics in patterning precision appears unlikely.

## Editor's evaluation

The authors use analytic calculations and numerical simulations to convincingly show that the purported benefits of nonlinear decay in morphogen gradients may be marginal in some cases and completely reversed in others (far from the concentration source). This is a valuable contribution to the field, as it questions common assumptions about the biological function of non-linear morphogen decays during development.

**\*For correspondence:**
dagmar.iber@bsse.ethz.ch

**Competing interest:** The authors declare that no competing interests exist.

## Introduction

According to Wolpert's famous French flag model (*Wolpert, 1969*), morphogen gradients encode readout positions $x_\theta$ via concentration thresholds $C_\theta = C(x_\theta)$, and differentiating cells base their fate decisions on whether the local morphogen concentration lies above or below such thresholds (*Figure 1A*). Thus, these readout positions mark the boundary locations between domains of different cell fates. Variations in the morphogen profile result in variations in the readout positions. The accuracy of the spatial information carried by morphogen gradients can be quantified with the positional error, which is defined as the standard deviation of the readout positions over different gradient realizations (*Vetter and Iber, 2022*):

$$\sigma_x = \text{stddev}\left[x_\theta\right].$$

(1)

How the observed precision of tissue patterns arising from this principle is achieved, in spite of natural molecular noise in morphogen production, transport, decay, internalization, turnover and other sources of variability, is a key question in developmental biology (*Lander, 2011*; *Vetter and Iber, 2022*; *Iber and Vetter, 2022*).

Morphogen dynamics are often described by reaction-diffusion equations of the form (*Lander et al., 2009*).

$$\frac{\partial C}{\partial t} = D\Delta C - dC^n/C_{\text{ref}}^{n-1}$$

(2)

**Figure 1.** Comparison of linear and non-linear morphogen gradients. (**A**) According to the French flag model, morphogen gradients provide the spatial information required for tissue patterning via concentration thresholds $C_\theta$, numbered by $\theta = 1, 2, 3$ etc. If a cell lies above or below a certain threshold $C_\theta$, it switches fate, resulting in domain boundaries forming at the respective cell borders at $x = x_\theta$ (blue and red lines). The morphogen source is located at $x = x_0 = 0$. (**B**) Linear decay leads to exponential gradients. Changes in the gradient amplitude $C_0$ (different colours) lead to a shift $\Delta x$ that is independent of the amplitude. (**C**) Non-linear decay ($n = 2$) leads to power-law gradients. The shift $\Delta x$ due to a change of $C_0$ is amplitude-dependent. (**D, E**) Noisy example gradients obtained numerically. Cell boundaries are denoted by black ticks along the patterning axis. Molecular kinetic noise and cell area variability leads to noisy gradients. For a fixed readout threshold $C_\theta$, variable gradients result in different readout positions $x_{\theta,j}$ (inset plots). Non-linear decay (**E**) leads to shallower gradients further in the patterning domain compared to linear decay (**D**).

with morphogen concentration $C$, diffusion coefficient $D$, and decay rate $d$. $C_{ref}$ is a constant reference concentration that we introduce to make all units independent of $n$. The exponent $n$ models linear ($n = 1$) or non-linear ($n > 1$) decay of the morphogen. Linear decay leads to exponential gradient profiles (*Figure 1B*) of the form (*Lander et al., 2002*).

$$C(x) = C_0 e^{-x/\lambda}, \qquad \lambda = \sqrt{\frac{D}{d}} \tag{3}$$

with an amplitude $C_0$ at the source at $x = 0$, and a characteristic decay length $\lambda$, set by the diffusion coefficient $D$ and the degradation rate $d$. Non-linear decay, on the other hand, results in shifted power-law gradients (*Figure 1C*; *Eldar et al., 2003*)

$$C(x) = C_0 \left(1 + \frac{x}{m\lambda_m}\right)^{-m}, \qquad m = \frac{2}{n-1} \tag{4}$$

where $\lambda_m$ is a gradient length scale that depends on $D/d$, $n$ and $C_0/C_{\mathrm{ref}}$ (see Appendix 1). Non-linear decay would for instance arise in case of cell lineage transport, when ligands interact with receptor clusters, or if ligand binding results in receptor upregulation, as is the case for several morphogens, most prominently for Hedgehog (Hh) (*Eldar et al., 2003*; *Nahmad and Stathopoulos, 2009*; *Wartlick et al., 2009*; *Balaskas et al., 2012*). Most reported morphogen gradient profiles have been fitted assuming linear decay ($n = 1$) (*Gregor et al., 2007*; *Gregor et al., 2008*; *Kicheva et al., 2007*; *Wartlick et al., 2011*; *Wartlick et al., 2014*; *Cohen et al., 2015*; *Zagorski et al., 2017*; *Mateus et al., 2020*). For the FGF8 gradient in the developing mouse brain, $n \approx 4$ has been reported (*Chan et al., 2017*).

Embryos are subject to molecular noise, which can cause fluctuations in morphogen production and transport rates, and consequently, in the gradient amplitudes and morphogen fluxes from the source to the patterned cells. This results in shifts $\Delta x$ between different gradient realisations (*Figure 1B and C*). In the case of linear decay, the shift is only related to the *relative* morphogen influx or amplitude and not to the *absolute* morphogen levels (Appendix 1, *Equation 11*). However, with non-linear decay, the shift depends on the absolute levels, with higher influxes resulting in smaller shifts (Appendix 1, *Equation 12*). Previous research (*Eldar et al., 2003*) suggested that the circumstance that this shift vanishes for power-law gradients at sufficiently large morphogen influx values leads to more robust patterning, because the readout position becomes independent of the influx in this limit when the morphogen decay is non-linear (Appendix 1). In other words, if two tissues are patterned by two different noise-free power-law gradients, both with high (but different) morphogen influxes from the source, the resulting gradients will be nearly identical, resulting in a reproducible pattern. For exponential gradients, the shift will not disappear, and the gradients will thus differ. However, the gradients that result from non-linear decay also possess significantly shallower tails, relative to the higher concentration (*Figure 1C*). Their usefulness for patterning has therefore been questioned (*Lander et al., 2009*), and it has remained unclear whether nonlinearity in the morphogen decay would in fact help achieving higher positional accuracy. Indeed, to first order, the positional error of variable gradients is inversely proportional to the magnitude of their slope (*Gregor et al., 2007*; *Vetter and Iber, 2022*) according to

$$\sigma_x \approx \left|\frac{\partial C}{\partial x}\right|^{-1} \sigma_C, \tag{5}$$

where $\sigma_C$ is the standard deviation of local morphogen concentration and $x$ denotes the patterning axis. This suggests that for patterning precision, the benefit of a smaller positional shift of gradients with $n > 1$ might be offset or even overcompensated by their flatter shape further from the source.

In the previous analysis (*Eldar et al., 2003*), molecular noise was considered only in the form of a fold-change in the morphogen amplitude or influx between different gradient realisations, resulting in shifted deterministic gradients, as shown in *Figure 1B and C*. To account for the intrinsic stochasticity of biological systems, we now extend this deterministic view by incorporating kinetic variability into the model, as depicted in *Figure 1D and E*. This is achieved by introducing randomness into all kinetic parameters of the reaction-diffusion equation. Our model is cell-based, meaning that each cell in the tissue is assigned its own specific variable kinetic parameters, emulating inter-cellular variability (Methods). With this quantitative statistical tool, we demonstrate numerically that the positional error

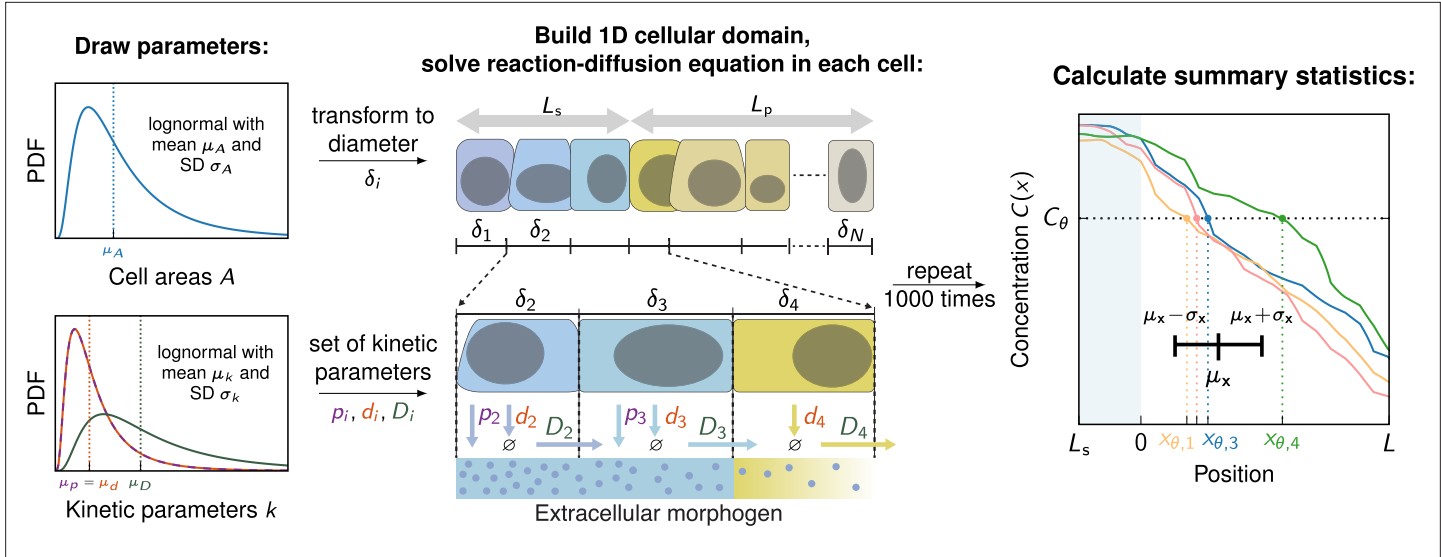

**Figure 2.** Numerical model to simulate noisy gradients. A 1D cellular domain is constructed by drawing cell areas from log-normal distributions with mean cell area $\mu_A$ and standard deviation $\sigma_A$. Cell areas are then converted to diameters ($\delta_i$). This procedure is repeated $N$ times until source and patterning domains of length $L_s$ and $L_p$ are filled with cells. Kinetic parameters $k = p, d, D$ are drawn independently from log-normal distributions with a mean $\mu_k$ and standard deviation $\sigma_k$ for each cell. Production only takes place in the source (blue cells). Then, the reaction-diffusion equation (**Equation 6**) is solved on the cellular domain, generating one noisy gradient $C_j(x)$. To determine a unique readout concentration of a cell, the average concentration along the cell boundary is computed for each cell in the patterning domain. Based on these concentrations the readout position $x_{\theta,j}$ where $C_j(x_{\theta,j}) = C_\theta$ is recorded for each gradient. This step is repeated 1000 times. Lastly, the average readout position $\mu_x$ and the positional error $\sigma_x$ is calculated based on the 1000 noisy gradients. PDF denotes the probability density function.

of noisy morphogen gradients does not significantly improve with non-linear decay. In the contrary, if the morphogen cannot leave the patterned tissue opposite of the source, the power-law gradients become shallow in a substantial part of the domain, leading to reduced positional accuracy with non-linear decay.

## Results

### Noisy gradient model

We simulated steady-state diffusion to study the impact of non-linear decay on the precision of noisy morphogen gradients. Our model uses a one-dimensional cellular domain composed of a source of length $L_s$ and a patterning region of length $L_p$ (**Figure 1D and E**). To represent morphogen-secreting source cells explicitly, the diffusion equation (**Equation 2**) was extended by a morphogen production term, resulting in

$$0 = D\frac{\partial^2 C}{\partial x^2} - dC^n/C_{\text{ref}}^{n-1} + pH(-x). \qquad (6)$$

Here, $H(x)$ is the Heaviside function, ensuring that production at rate $p$ only occurs in the source ($x < 0$). Zero-flux boundary conditions were used at both outer ends of the tissue, mimicking a situation in which morphogen molecules are restricted to the patterned tissue by an impermeable boundary:

$$\frac{\partial C}{\partial x}(-L_s) = 0 = \frac{\partial C}{\partial x}(L_p).$$

We generated variable morphogen gradients by numerically solving **Equation 6** with kinetic parameters $p_i$, $d_i$ and $D_i$, and cell areas $A_i$ independently drawn from log-normal distributions for each cell $i = 1, ..., N$ in the domain (**Vetter and Iber, 2022**; **Adelmann et al., 2022**; **Figure 2**, for details see Methods). The individual gradient realisations $C_j(x)$ can be thought of as representing different embryos, denoted by the index $j$. They exhibit inter- and intra-tissue variability due to the stochastic nature of the three kinetic parameters that vary from cell to cell. Cells have to convert the

spatial morphogen distribution they are exposed to into a single concentration value, which determines their fate in the tissue according to the French flag model. There are several ways cells may achieve this, such as averaging the morphogen signal over their entire cell surface, beyond their cell surface via a cilium, or reading out the signal at a single point. In a recent study, we found that the readout mechanism has little impact on the gradient precision perceived by the cells (*Adelmann et al., 2022*). We therefore only analysed the case where cells average the morphogen signal over their cell surface, or over their diameter in the 1D model here, respectively. The concentration in each cell is then compared to the threshold concentration $C_\theta$ and the location $x_{\theta,j}$ of the first cell whose sensed concentration subceeds this threshold is recorded. This process is repeated for all gradients $j$, allowing to compute the positional error according to its definition (*Equation 1*), $\sigma_x = \text{stddev}_j\{x_{\theta,j}\}$ to quantify the precision of the positional information conveyed by the morphogen gradients.

## Model parameters

To define the stochastic nature of the morphogen kinetics involved in the formation of the gradients, we express the mean value and standard deviation of a parameter $q$ by $\mu_q$ and $\sigma_q$, respectively (*Figure 2*). Based on measurements of the Hedgehog morphogen gradient in the *Drosophila* wing disc and mouse neural tube (*Kicheva et al., 2007*; *Cohen et al., 2015*; *Vetter and Iber, 2022*), we used a mean diffusivity of $\mu_D = 0.033$ μm²/s and a mean gradient length $\mu_\lambda = 20$ μm. We furthermore set the average degradation rate to $\mu_d = \mu_D/\mu_\lambda^2$, and the average production rate to $\mu_p = \mu_d C_{\text{ref}}$, where $C_{\text{ref}} = 1$ arb. units to normalise the concentrations. Other specific values of gradient parameters would not change the results reported here, which are for the steady state, but would only alter the timescale it takes for the steady state to be reached. The noise-to-signal ratio in each quantity $q$ is given by the corresponding coefficient of variation, $\text{CV}_q = \sigma_q/\mu_q$. Reported physiological noise levels in morphogen production, decay, and transport differ between morphogens and tissues, but are around $\text{CV}_{p,d,D} \approx 0.3$ (*Vetter and Iber, 2022*), which we use to define the distribution widths of the kinetic parameters.

In addition to the morphogen kinetics, our simulations also include cell-to-cell variability in the cell areas. The widths and cross-sectional areas of cells vary in all layers along the apical-basal axis (*Gómez et al., 2021*). Most quantifications have been carried out on the apical surface. One of the highest reported values for the apical area CV is found in the vertebrate neural tube ($\text{CV}_A \approx 0.9$) (*Escudero et al., 2011*; *Guerrero et al., 2019*; *Bocanegra-Moreno et al., 2022*), but most values are considerably lower (*Kokic et al., 2019*). We therefore used $\text{CV}_A = 0.5$ in all simulations unless specified otherwise.

The diffusion coefficient, $D$, and the degradation rate, $d$, set the steady-state patterning length scale, $\lambda = \sqrt{D/d}$. Thus, our results are independent of the absolute values chosen for $D$ and $d$, and only depend on their ratio. Positional quantities, such as the positional error, are reported relative to the average cell diameter, which in turn was chosen to be a fixed multiple of the average gradient decay length. We fixed the average cell diameter at a fourth of the exponential gradient length, $\mu_\delta/\mu_\lambda = 1/4$, as found in the developing mouse neural tube (*Kicheva et al., 2014*; *Cohen et al., 2015*).

## Impact of non-linear decay on gradient precision

We previously showed that in case of linear decay, there is a negligible impact of cell area variability as long as $\text{CV}_A < 1$ (*Adelmann et al., 2022*). We now find that this holds similarly for non-linear decay (*Figure 3A*), justifying the use of a fixed $\text{CV}_A = 0.5$ in the remainder of our analysis.

Much as for linear decay (*Adelmann et al., 2022*), the positional error scales with the square root of the mean cell diameter also for non-linear decay (*Figure 3B*). Small cell diameters, as observed in all known tissues that employ gradient-based patterning (*Adelmann et al., 2022*), are therefore important for high spatial precision also in case of non-linear decay.

The positional error increases from less than one cell diameter close to the source to about two cell diameters at a distance of 75 cell diameters away from the source (*Figure 3C*). Close to the distant domain boundary opposite of the source, where a no-flux condition was imposed, the positional error rapidly increases for non-linear decay, while remaining relatively low for linear decay. If only the production in the source is varied ($\text{CV}_p = 0.3$, $\text{CV}_{d,D} = 0$), the positional error remains constant as the readout distance from the source increases, but increases again sharply close to the distant end in case of non-linear decay (*Figure 3D*). But even for strong non-linearity ($n = 4$), the positional error

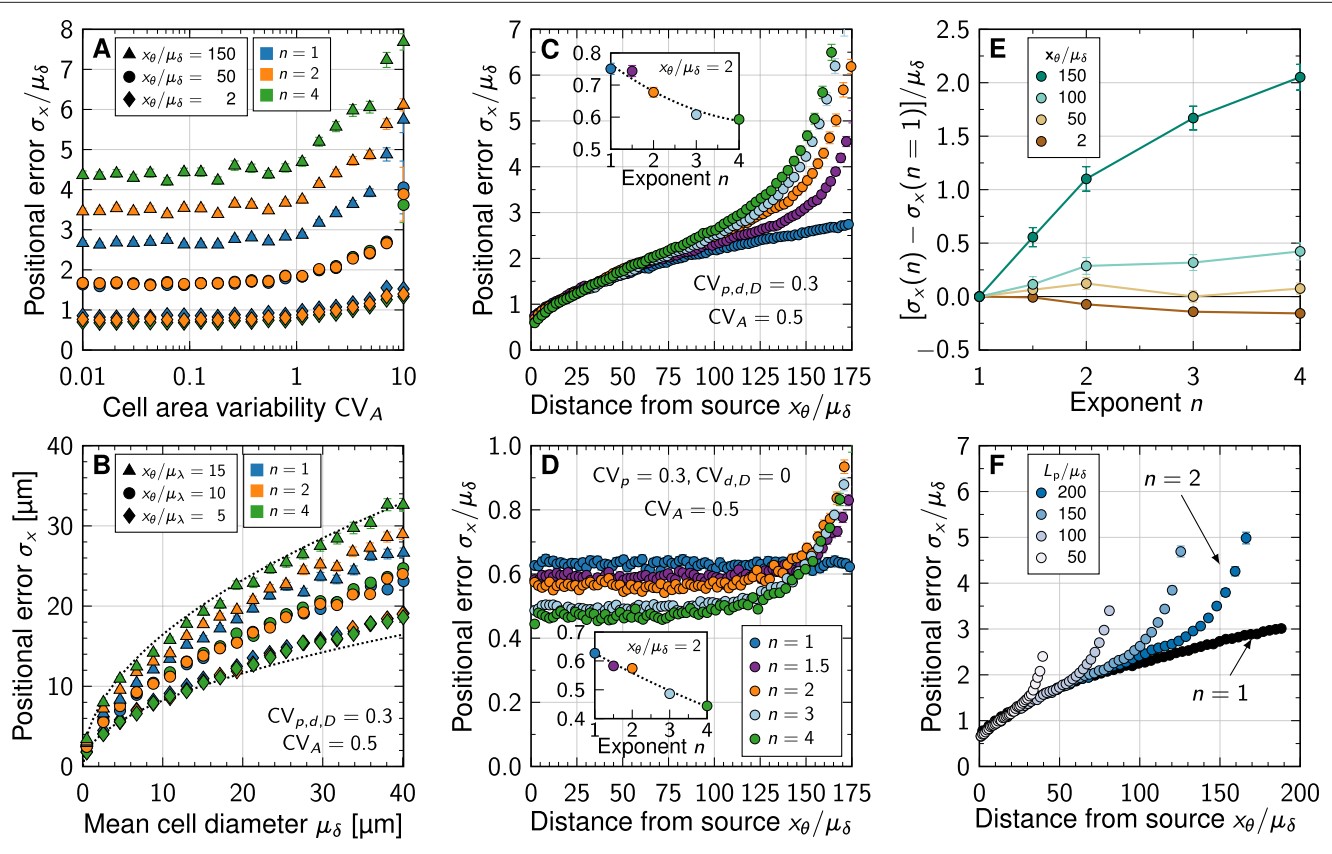

**Figure 3.** Impact of non-linear decay on gradient precision. (**A**) Physiological variability in the cross-sectional cell areas has no significant impact on gradient precision. The positional error $\sigma_x$ is plotted in units of the mean cell diameter $\mu_\delta$ at different readout positions in the patterning domain (symbols), and for different degrees of non-linearity (colours). (**B**) The positional error increases with the square root of the cell diameter, irrespective of $n$. Dotted lines show $\sigma_x = \alpha\sqrt{\mu_\delta}$ for $\alpha = 2.6, 5.2$ for reference, with lengths in units of µm. $L_p = 100\mu_\delta$. (**C**) Non-linear decay leads to a marginally lower positional error close to the morphogen source. Inset plot shows $\sigma_x/\mu_\delta$ at a distance of two cells from the source as a function of decay non-linearity. With a no-flux boundary at $x = L_p$, the shallowness of gradients from non-linear decay lets the positional error increase strongly far from the source. Colours correspond to different decay exponents $n$, as specified in panel D. (**D**) Variability in the production rate alone has no effect on the positional error along the domain for linear decay (blue). The stronger the non-linearity, the smaller the positional error close to the source (inset). Far from the source, the positional error increases rapidly with non-linear decay. (**E**) Difference between the positional error for $n > 1$ and for $n = 1$ relative to the mean cell diameter, at fixed readout positions (colours). (**F**) Effect of finite patterning domain size. The positional error increases close to the distant zero-flux boundary in case of non-linear decay (shades of blue, $n = 2$). Patterning remains precise across a larger distance in the case of linear decay (black, $n = 1$). In all panels, each data point represents the mean from $10^3$ independent simulations. Error bars represent standard errors.

remains in the sub-cellular range when only production noise is considered, as long as the readout position is further than about $\lambda$ away from the distal end.

Independent of whether all parameters are varied or only the production rate, the positional error drops in close vicinity to the source with stronger non-linearity in the decay (insets of **Figure 3C and D**). However, with less than 20% of a single cell diameter from $n = 1$ to $n = 4$, the effect is likely too small to be physiologically relevant. Further away from the source, linear decay yields a smaller positional error than non-linear decay (**Figure 3D–E**). No matter how long the patterning domain is, non-linearity always increases the positional error as the distal tissue boundary is approached (**Figure 3F**).

What then causes the increased positional errors with non-linear decay near the distal domain boundary? A zero-flux boundary condition there implies shallower gradients than on infinite domains: $C'(x) \to 0$ as $x \to L_p$. This effect occurs irrespective of $n$, but the spatial range over which the gradient flattens (and thus deviates from the pure exponential and shifted power-law forms for infinite domains, **Equations 3; 4**) increases with $n$. By virtue of **Equation 5**, non-linear decay thus leads to greater positional errors at readout positions in the vicinity of the distal boundary compared to linear decay.

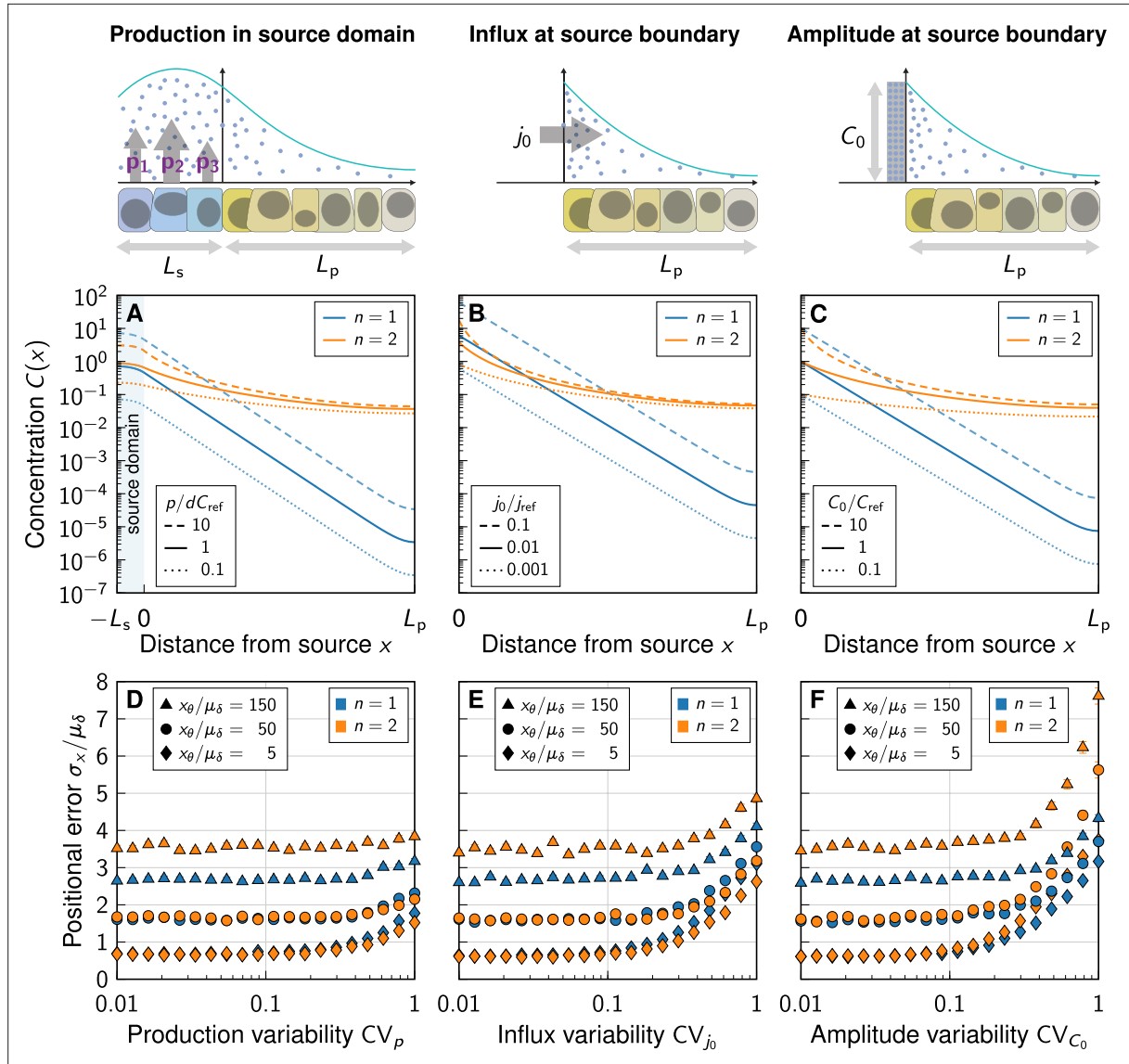

**Figure 4.** Impact of the boundary condition (BC) at the source. (**A–C**) Noise-free gradient shapes when the morphogen is either secreted in a source domain at rate $p$ (*Equation 6*) (**A**), with flux BC, $-D\partial C/\partial x|_{x=0} = j_0$ (**B**), or Dirichlet BC, $C(0) = C_0$ (**C**). No-flux BC were imposed at at the far end of the tissue (at $x = L_p$). (**D–F**) Positional error as a function of morphogen abundance variability, at different readout positions (symbols) and degrees of non-linearity (colours). Greater variability in the morphogen production rate (**D**), influx (**E**), and gradient amplitude (**F**) leads to a larger positional error above a certain threshold variability $\mathrm{CV} \gtrsim 0.1 - 0.3$. Kinetic variability was fixed at $\mathrm{CV}_{d,D} = 0.3$ (except for $\mathrm{CV}_P$ in **D**). Further parameters: $\mu_{j_0} = \mu_D C_{ref}/\mu_\lambda$ (**E**), $\mu_{C_0} = C_{ref}$ (**F**). In panels **D–F**, each data point represents the mean from $10^3$ independent simulations. Error bars represent standard errors.

In summary, our computer simulations of noisy morphogen gradients suggest that it is insufficient to quantify gradient robustness and patterning precision by considering variability in the morphogen production alone. Moreover, if the morphogen cannot exit the patterning domain opposite of the source, shifted power-law gradients that result from non-linear morphogen decay flatten over a significantly larger range than exponential gradients, leading to increased positional errors. The gain in positional accuracy close to the source for non-linear decay is negligible and therefore barely physiologically relevant. Overall, exponential gradients lead to more robust patterning.

## Impact of boundary condition at the source

Given the impact of the distal domain boundary, we wondered whether the representation of the morphogen source by either a spatial production domain (*Figure 4A*), by a flux boundary condition $-DC'(0) = j_0$ (*Figure 4B*) as used by *Eldar et al., 2003*, or by a fixed gradient amplitude $C(0) = C_0$

(*Figure 4C*) would affect the positional error predicted by the model. While there are small quantitative differences, the gradient shapes (*Figure 4A–C*) and positional errors (*Figure 4D–F*) are overall very similar.

As we increase the variability in the production rate via $CV_p$ (*Figure 4D*), in the influx from the source via $CV_{j_0}$ (*Figure 4E*), or in the gradient amplitude at the source boundary via $CV_{C_0}$ (*Figure 4F*), we find the smallest increase in the positional error for the production rate and the largest increase for the gradient amplitude. Neumann or Dirichlet boundary conditions thus overestimate the positional error when the variability in the source is high. Instead of using such boundary conditions, a spatial source domain should explicitly be modeled, where applicable. With the physiological values $CV_p \approx 0.3$ and $CV_{C_0} \lesssim 0.3$ (*Vetter and Iber, 2022*), however, variability in the morphogen production plays merely a subordinate to moderate role in the overall gradient variability. Molecular noise in morphogen degradation and diffusivity dominates the patterning precision.

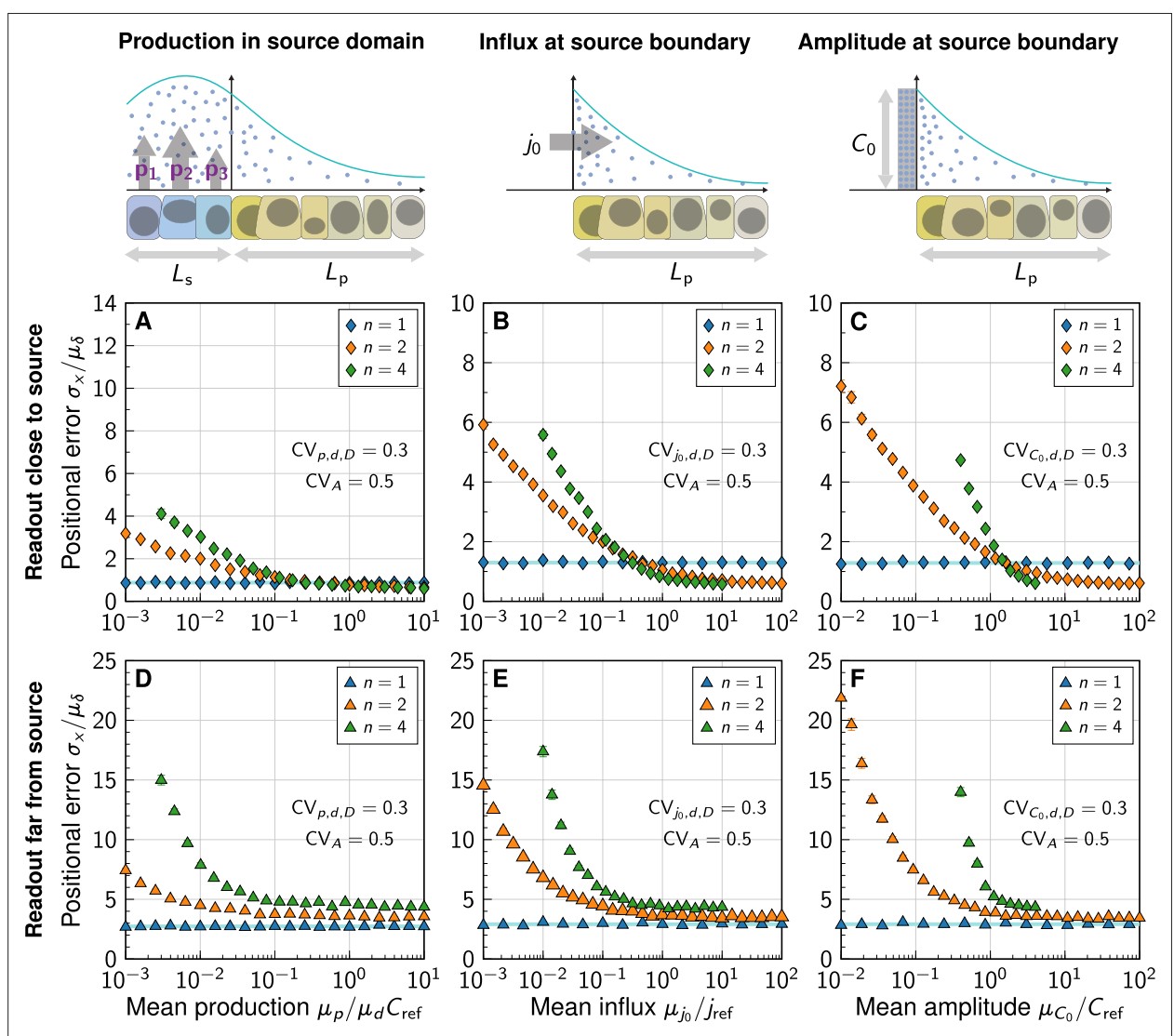

**Figure 5.** Impact of the morphogen source strength. Numerically obtained spatial patterning accuracy in units of average cell diameters $\mu_\delta$ at different positions in the tissue (symbols) and for different degrees of non-linearity (colours). (**A–C**) Readout close to the source, at $x_\theta = 5\mu_\delta$; (**D–F**) Readout far from the source, at $x_\theta = 150\mu_\delta$. Morphogen production scenarios are identical to *Figure 4*: Production in a source domain with morphogen-secreting cells (**A,D**), with a morphogen influx from the source at the source boundary (**B,E**), and with a specified morphogen concentration at the source boundary (**C,F**). Very low (high) influxes or amplitudes lead to flat (steep) gradients at strong decay non-linearity, limiting the parameter range over which the positional error can be reliably determined for $n = 4$ (**B,C,E,F**). In all panels, each data point represents the mean from $10^3$ independent simulations, error bars represent standard errors.

## Impact of the morphogen source strength

As the gradient amplitude determines the sensitivity of the readout position to amplitude changes for non-linear decay (Appendix 1, *Equation 12*) but not for linear decay (Appendix 1, *Equation 11*), the average morphogen production rate is expected to affect the patterning accuracy in the case of non-linear decay, but not for linear decay. We put this theoretical prediction to the test by varying the mean relative production rate, the mean influx from the source, and the mean morphogen amplitude in the three different simulated morphogen production models. Changes in these parameters have no effect on the positional error if morphogen degradation is linear, which is consistent with the theory (*Figure 5A–F*, blue lines). With non-linear decay, on the other hand, we indeed observe the positional error to be highly dependent on morphogen abundance. Precision arguments previously brought forward for deterministic morphogen gradients (*Eldar et al., 2003*) do not appear to directly quantitatively translate to the positional error in settings where cell-to-cell variability is included, and where morphogen production remains at physiological levels.

For high morphogen supply levels, non-linear decay leads to a smaller positional error close to the source (*Figure 5A–C*). The effect is, however, substantially less pronounced in the model that includes a spatial morphogen source domain (*Figure 5A*) than in those that do not (*Figure 5B and C*), highlighting once again the limitations of the latter. With an explicit source domain, non-linear decay yields only marginally more spatial accuracy, when production is high ($p/dC_{\text{ref}} \gtrsim 0.4$). Lower production levels increase the positional error close to the source substantially in all three models, reaching several cell diameters, for $n > 1$. The gradients effectively flatten out at low production, reducing their usefulness for spatial tissue patterning. The stronger the non-linearity in the degradation, the more pronounced this loss of patterning precision.

Further away from the source, the benefit of non-linear decay is lost entirely, and exponential gradients remain more precise than shifted power-law gradients also at high morphogen supply levels (*Figure 5D–F*).

In summary, simplified models without explicit representation of morphogen-secreting cells overestimate the beneficial impact of non-linear decay on patterning precision. In all models considered here, the benefit of non-linear morphogen decay is restricted to a close vicinity of the morphogen source, where patterning precision is high anyway (*Vetter and Iber, 2022*) and may thus not be as critical for robust development, and to a regime of very strong morphogen production. Further into the tissue, and at moderate morphogen abundance, linear decay yields more accurate patterning.

## Discussion

Non-linear morphogen decay was proposed as a potential precision-enhancing mechanism for tissue patterning in the seminal theoretical work by *Eldar et al., 2003* in a deterministic setting, where morphogen gradients are devoid of noise. Here we have explored this idea with a stochastic model, taking noisy gradients into account, as they arise from cell-to-cell variability in morphogen kinetics. The surprising outcome of our quantitative analysis is that, while a small advantageous effect indeed exists near the morphogen source, this gain is outweighed by a substantial loss of precision in the spatial information that signalling gradients provide to cells in the interior and distal parts of a patterned tissue when morphogen decay is non-linear. In tissues that pose a diffusion barrier to the signalling molecule at their boundary, shifted power-law gradients that emerge with self-enhanced degradation, flatten out over a substantial portion of the spatial domain, whereas exponential gradients remain more graded (*Figure 1*). This leads to greater spatial precision with linear decay (*Figure 3*), and is contrary to the original expectation (*Eldar et al., 2003*).

This long-range boundary effect is not the only reason why linear morphogen decay is favourable for precise pattern formation. The positional error, which is the decisive quantity that measures the spatial accuracy with which cells can determine their location in the pattern, and ultimately their fate in differentiation, is highly sensitive to morphogen supply levels when morphogen decay is non-linear, but largely insensitive when decay is linear (*Figure 5*). This implies that patterning is more robust to variations in the size and strength of the morphogen-secreting source, if decay is linear. These results challenge the established view that power-law gradients buffer fluctuations in morphogen production (*Eldar et al., 2003*). We find that the positional error behaves in the opposite way, buffering production fluctuations only with linear, but not with non-linear decay. From an evolutionary perspective, the

linear case may be favoured, as patterning precision is unaffected by changes in the size and kinetics of the morphogen-secreting source only if $n = 1$.

Our study demonstrates that a stochastic approach is required to quantify patterning precision of real noisy gradients. Moreover, we find the positional error to be overestimated in simplified models that replace the morphogen-secreting cells by a Neumann or Dirichlet boundary condition (*Figure 4*). Based on this, we recommend to include an explicit representation of the source in future theoretical or numerical work on the subject, as we did with *Equation 6*.

Distinguishing exponential gradients from shifted power laws can be very difficult in practice, as they can appear similar over the short distances over which they can be reliably measured with classical imaging techniques. The FGF8 gradient in the developing mouse brain is the only case we are aware of where $n > 1$ has been reported robustly (*Chan et al., 2017*), and whether this is linked to patterning precision in any way remains unclear. Available gradient data in other systems, such as Sonic Hedgehog and Bone Morphogenetic Protein in the neural tube (*Zagorski et al., 2017*), is too variable to confidently reject the hypothesis that $n = 1$. Most further reports of morphogen gradient shapes (*Gregor et al., 2007*; *Gregor et al., 2008*; *Kicheva et al., 2007*; *Wartlick et al., 2011*; *Wartlick et al., 2014*; *Cohen et al., 2015*; *Zagorski et al., 2017*; *Mateus et al., 2020*) are consistent with exponentials within measurement accuracy. New measurement techniques are needed to determine whether non-linear decay is at work in the formation of known morphogen gradients during development. In light of our findings, a physiological role of non-linear ligand decay in patterning precision appears implausible. If anything, our data suggest an overall advantage of linear decay, also considering the evolutionary aspect of tissue size and protein synthesis rate differences between species.

The morphogen concentration declines significantly over several orders of magnitude, independent of whether there is linear or non-linear decay. At low morphogen concentrations, thermal fluctuations and stochastic binding kinetics of ligands and receptors will affect gradient and readout precision (*Berg and Purcell, 1977*; *Lauffenburger and Linderman, 1996*; *Lander et al., 2009*). Cells can, in principle, achieve high readout precision despite such fluctuations via spatial and temporal averaging (*Lauffenburger and Linderman, 1996*). To assess such effects, quantitative measurements of morphogen numbers and cellular responses would be required far away from the source. This requires the further development of more sensitive measurement technology (*Lelek et al., 2021*). Once the absolute concentration levels of the morphogen gradients can be determined, it can be assessed whether the approximation of the gradients by a continuous functions is valid along the whole tissue or, whether discrete models have to be considered.

In future work, the simulated gradients can be used as inputs to complex downstream networks, and the effect of noise in the readout can be studied. However, these downstream networks would not alter the relative precision of gradients generated by linear and non-linear decay. In conclusion, non-linear decay may slightly enhance precision close to the source, but it rapidly deteriorates far from the source.

## Methods
### Generation of variable morphogen gradients
To generate noisy morphogen gradients numerically, we constructed the one-dimensional cellular domains in an iterative process, cell by cell. For each cell $i$, an area $A_i$ was drawn from a log-normal distribution with specified mean value $\mu_A$ and coefficient of variation $CV_A$ (*Adelmann et al., 2022*). The drawn area was then converted to a cell diameter $\delta_i = 2\sqrt{A_i}/\pi$. Using the transformation properties of log-normal distributions, the cell areas was drawn according to

$$\mu_A = \pi(\mu_\delta/2)^2(1 + CV_A^2)^{1/4},$$

allowing to accurately fix the mean cell diameter $\mu_\delta$. This procedure was repeated for cells $i = 1, 2, 3...$ until the sum of the diameters equaled the source length $L_s$ or the patterning domain length $L_p$. The spatial axis was then discretized into cellular sub-intervals accordingly (*Figure 2*). We used a patterning domain length of 200 cells ($L_p = 200\mu_\delta$) and a source domain length of 5 cells ($L_s = 5\mu_\delta$), unless otherwise stated.

Once the patterning axis was constructed, the kinetic parameters $p_i, d_i, D_i$, were drawn from log-normal distributions for each cell $i$ independently. For simulations without explicit source domain, a random morphogen influx $j_0$ or an amplitude $C_0$ was also drawn from a log-normal distribution. We then numerically solved *Equation 6* using Matlab's built-in fourth-order boundary value problem solver bvp4c (version R2020b). At cell boundaries, we imposed continuity of both the morphogen concentration and flux. Repeating this procedure $10^3$ times using independent random parameters and cell areas yielded statistically independent realisations of noisy morphogen gradients. To estimate the standard errors of the positional errors as shown in the plots, we used bootstrapping.

## Choice of parameter distribution

In this article, we assume log-normally distributed cell areas and kinetic parameters, analogous to our previous works (*Vetter and Iber, 2022*; *Adelmann et al., 2022*). For the cell areas, this choice is rooted in the reported distributions of apical areas in the *Drosophila* larval and prepupal wing discs, and in the mouse neural tube (*Sánchez-Gutiérrez et al., 2016*; *Guerrero et al., 2019*). The results reported here are, however, largely independent of the probability distribution, as long as it satisfies certain physiological criteria:

- The random parameters must be strictly positive. This rules out probability distributions which allow for negative values, including for example a normal distribution.
- The probability of drawing a near-zero parameter must vanish quickly. This is because tiny diffusion coefficients, fluxes, or amplitudes do not allow for successful patterning over biologically relevant distances or timescales. A normal distribution truncated at zero, for example, is ruled out because minuscule diffusion coefficients would occur frequently.

In recent related work (*Adelmann et al., 2022*), we demonstrated that other distributions which fulfill the above criteria yield similar results.

If the morphogen source is not modeled explicitly (omitting the production term in *Equation 6*), the gradient amplitude or morphogen influx levels at the source boundary can serve as a proxy for the production of the morphogen. For these simulations, the amplitudes and fluxes were also drawn from log-normal distributions. The width of these distributions is controlled via their coefficients of variation, $\mathrm{CV}_{C_0}$ or $\mathrm{CV}_{j_0}$ as specified in the respective figures.

## Acknowledgements

This work was partially funded by SNF Sinergia grant CRSII5_170930.

## Additional information

### Funding

| Funder | Grant reference number | Author |
|---|---|---|
| Schweizerischer Nationalfonds zur Förderung der Wissenschaftlichen Forschung | CRSII5_170930 | Roman Vetter<br>Dagmar Iber |

The funders had no role in study design, data collection and interpretation, or the decision to submit the work for publication.

### Author contributions

Jan Andreas Adelmann, Software, Formal analysis, Investigation, Visualization, Writing - original draft, Writing - review and editing; Roman Vetter, Software, Formal analysis, Supervision, Investigation, Visualization, Methodology, Writing - original draft, Writing - review and editing; Dagmar Iber, Conceptualization, Resources, Supervision, Funding acquisition, Methodology, Writing - original draft, Project administration, Writing - review and editing

## Author ORCIDs

Jan Andreas Adelmann  http://orcid.org/0000-0003-0876-3363
Roman Vetter  http://orcid.org/0000-0003-2901-7036
Dagmar Iber  http://orcid.org/0000-0001-8051-1035

## Decision letter and Author response

Decision letter https://doi.org/10.7554/eLife.84757.sa1
Author response https://doi.org/10.7554/eLife.84757.sa2

## Additional files

### Supplementary files

• MDAR checklist

### Data availability

The current manuscript is a computational study, so no data has been generated for this manuscript. The source code is released under the 3-clause BSD license. It is available as a public git repository at https://git.bsse.ethz.ch/iber/Publications/2022_adelmann_vetter_nonlinear_decay (copy archived at *Adelmann, 2023*).

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

## Appendix 1

### Qualitative difference between linear and non-linear morphogen decay

In this section we present some theoretical considerations about the consequences of nonlinear decay for noise-free morphogen gradients. We consider deterministic steady-state gradients obtained by analytically solving

$$\frac{\partial C}{\partial t} = 0 = D\Delta C - dC^n/C_{\text{ref}}^{n-1}. \tag{7}$$

Solving *Equation 7* on an infinite one-dimensional domain for linear morphogen decay ($n = 1$) with a concentration that drops to zero at infinite distance from the source ($C(x) \to 0$ as $x \to \infty$), results in exponential gradient profiles (*Figure 1A*) of the form

$$C(x) = C_0 e^{-x/\lambda}, \qquad \lambda = \sqrt{\frac{D}{d}}, \tag{8}$$

with an amplitude $C_0$ at the source at $x = 0$. The amplitude can be set by Dirichlet boundary conditions or by flux boundary conditions at the source, $-D\partial C/\partial x|_{x=0} = j_0$. Imposing flux boundary conditions leads to an amplitude $C_0 = j_0\lambda/D$. Thus, with linear decay, influx and amplitude are proportional.

Non-linear decay (*Equation 7*, $n > 1$), results in shifted power-law gradients (*Figure 1A*) that can be expressed as

$$C(x) = C_0 \left(1 + \frac{x}{m\lambda_m}\right)^{-m}, \qquad m = \frac{2}{n-1}, \tag{9}$$

where

$$\lambda_m = \lambda\sqrt{1 + \frac{1}{m}}\left(\frac{C_{\text{ref}}}{C_0}\right)^{\frac{1}{m}} \tag{10}$$

is a length scale determining the shift in the power law, and $C_0 = C(0)$ is the amplitude analogous to *Equation 8*. As the linear decay is approached ($n \to 1$), $m$ diverges ($m \to \infty$), the power-law length scale approaches the exponential length scale ($\lambda_m \to \lambda$), and the power-law gradients (*Equation 9*) become exponential (*Equation 8*). For a flux boundary condition at the source, the morphogen amplitude is

$$C_0 = \frac{j_0\lambda_m}{D} = \left(\lambda\sqrt{1 + \frac{1}{m}}\frac{j_0}{D}C_{\text{ref}}^{\frac{1}{m}}\right)^{\frac{m}{m+1}}.$$

Amplitude and influx at the source boundary are thus not proportional for non-linear morphogen decay, unlike in the linear case. Moreover, power-law gradients do not possess a constant gradient decay length $\lambda$ that quantifies a distance over which a fold-change in morphogen concentration occurs. Nevertheless, if one were to locally fit an exponential to the power-law gradient (*Eldar et al., 2003*),

$$x^{-m} \sim \exp[-x/\lambda(x)],$$

one would observe the "gradient decay length" $\lambda(x)$ to increase with the distance from the source according to $\lambda(x) = x/(m\ln x)$ (*Figure 1C*).

Morphogen gradients define readout positions $x_\theta$ via concentration thresholds $C_\theta = C(x_\theta)$ (*Figure 1A, D and E*). For linear decay, the readout position follows from *Equation 8* as

$$x_\theta = \lambda\ln\frac{C_0}{C_\theta},$$

and for non-linear decay from *Equation 9* as

$$x_\theta = m\lambda_m\left(\left(\frac{C_0}{C_\theta}\right)^{\frac{1}{m}} - 1\right).$$

Due to inevitable molecular noise in morphogen production, transport, and decay, morphogen gradients differ between embryos, and hence readout positions $x_{\theta,i}$ vary between different gradient realisations $i$ for both linear and non-linear morphogen decay (*Vetter and Iber, 2022*). In the past, the impact of changes in morphogen production on readout precision has been studied for gradients that remain otherwise unchanged between embryos (*Eldar et al., 2003*). We now revisit this scenario. In response to a change in the morphogen amplitude from $C_0$ to $C_0^*$, the readout position shifts along the patterning axis (*Figure 1B and C*). For linear decay, this shift $\Delta x$ is independent of the absolute gradient amplitude $C_0$ and depends only on the relative amplitude change, $C_0^*/C_0$, and the characteristic gradient length $\lambda$:

$$\Delta x = x_\theta^* - x_\theta = \lambda \ln \frac{C_0^*}{C_0}. \tag{11}$$

For non-linear decay, the shift is given by

$$\Delta x = m\lambda_m \left( 1 - \left( \frac{C_0}{C_0^*} \right)^{\frac{1}{m}} \right). \tag{12}$$

According to *Equation 12*, the shift $\Delta x$ is proportional to $\lambda_m$ which in turn is proportional to $C_0^{-1/m}$, implying that the shift increases with decreasing amplitude (Appendix 1, *Figure 1A,B*). This dependency of non-linear decay on the gradient amplitude qualitatively distinguishes linear from non-linear decay. Alternatively, the shift may be expressed as a function of a change in morphogen influx from the source from $j_0$ to $j_0^*$. For linear decay, it simply reads

$$\Delta x = \lambda \ln \frac{j_0^*}{j_0},$$

because flux and amplitude are proportional, making the shift again independent of absolute morphogen levels. For non-linear decay, however, amplitude and influx are related as

$$\left( \frac{C_0}{C_0^*} \right)^{\frac{1}{m}} = \left( \frac{j_0}{j_0^*} \right)^{\frac{1}{m+1}}.$$

The resulting readout shift is therefore

$$\Delta x = m\lambda_m \left( 1 - \left( \frac{j_0}{j_0^*} \right)^{\frac{1}{m+1}} \right)$$

with a power-law length scale $\lambda_m$ that can be expressed in terms of the influx $j_0$ as

$$\lambda_m = \lambda \left( \sqrt{1 + \frac{1}{m}} \left( \frac{j_{\text{ref}}}{j_0} \right)^{\frac{1}{m}} \right)^{\frac{m}{m+1}}, \qquad j_{\text{ref}} = \frac{DC_{\text{ref}}}{\lambda}.$$

Therefore, since $\Delta x$ is proportional to $\lambda_m$, which is in turn proportional to $j_0^{-1/(m+1)}$, the shift also increases with decreasing influx (Appendix 1 *Figure 1A,C*), albeit slower than with the amplitude.

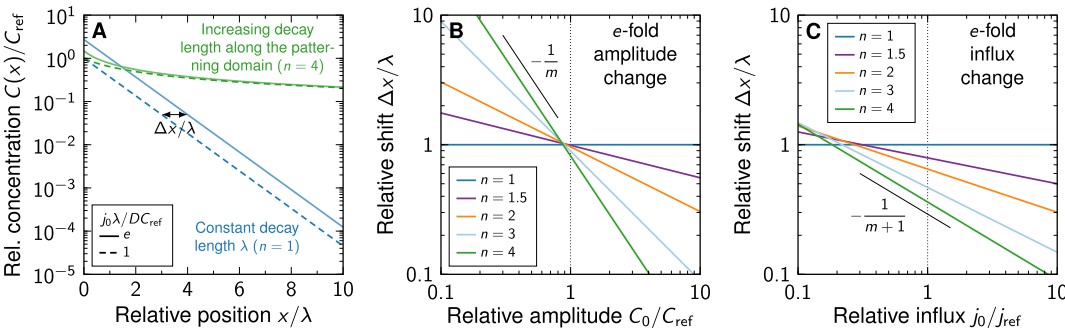

**Appendix 1—figure 1.** Shift in morphogen gradients due to changes in morphogen production. (**A**) Comparison of noise-free gradients arising from linear (blue) and non-linear (green) decay. A fold-change in the influx $j_0$ from the source shifts the gradients by $\Delta x$. (**B**) Positional shift of the morphogen gradient as a function of the amplitude and degree of non-linearity, for a fold-change in the amplitude, $C_0^*/C_0 = e$. (**C**) Positional shift as a function of the influx and degree of non-linearity, for a fold-change in the influx, $j_0^*/j_0 = e$.

