## [Editor Report]

The authors use analytic calculations and numerical simulations to convincingly show that the purported benefits of nonlinear decay in morphogen gradients may be marginal in some cases and completely reversed in others (far from the concentration source). This is a valuable contribution to the field, as it questions common assumptions about the biological function of non-linear morphogen decays during development.

---

## [Decision Letter]

**Decision letter after peer review:**

Thank you for submitting your article "Patterning precision under non-linear morphogen decay and molecular noise" for consideration by *eLife*. Your article has been reviewed by 3 peer reviewers, including Gordon J Berman as the Reviewing Editor and Reviewer #1, and the evaluation has been overseen by Aleksandra Walczak as the Senior Editor.

Essential revisions:

Although the reviewers all agreed that the study was rigorously performed and reaches an interesting and valuable conclusion, the paper is difficult read for people unfamiliar with the current literature. The reviewers ask that the authors clarify the text of the article (see reviews below), allowing a reader uninformed in the field of morphogen gradient calculations (e.g., a biophysicist working in another field) to have an easier time understanding the key findings here.

*Reviewer #1 (Recommendations for the authors):*

These morphogens often serve as transcription factors to help set up other, also noisy, downstream morphogen gradients. Thus, it would be interesting to have the authors' comment on whether having a linear + a non-linear or a non-linear + a non-linear set of decay functionals might change the readout accuracy further away from the source. As these calculations might be time-consuming, I'm not specifically asking for them to be done, but it would be good to get the authors' take on the matter for the Discussion section.

My main concern with the submission is that the article is written in a very short format that is very difficult for someone outside the field (myself included) to read. All of the information was there and I was able to figure things out, but only with significant effort. I would recommend expanding the text somewhat and providing a bit more detail as to where previous mathematical approaches ended and where this one starts.

*Reviewer #2 (Recommendations for the authors):*

1. This is a bit of author preference, but I find that I prefer a free-standing methods section as opposed to having them interleaved into the results, as they are now. I think the free-standing section allows for a bit more elaboration on details without detracting from the narrative.

2. I generally found this paper a nice exploration of the question of how non-linear decay affects patterning. The paper is written in such a way that it may be challenging to read for those less familiar with the equations and conventions of mathematical modeling. If the authors wish to make the paper more accessible, I would suggest (in addition to the separation of methods) spending a bit more time defining the variables and parameters used throughout the paper, the justification for the different types of x/y axes used in the different graphs, and perhaps adding a cartoon or two illustrating a hypothetical tissue being patterned by a morphogen with some of the parameters/variables depicted. In addition, giving specific biological examples that correspond to some of the model considerations (e.g. boundary conditions) may be interesting for some readers.

3. It might be interesting to add two points to the Discussion section. First, the simulations here explore the effect of parameter variation, which might be a good representation of, e.g. how genetic variability between individuals affects parameters and therefore patterning. However, even with consistent mean parameter values, molecular stochasticity can also impact these processes – is this worth exploring in future work? Second, given that most (all?) morphogens seem to operate in more complex regulatory networks, with features like opposing gradients or mutual inhibition of morphogen targets, might the interaction between non-linear decay and these other features be worth exploring in the future?

*Reviewer #3 (Recommendations for the authors):*

I find that this paper is very hard to read. The results of this paper are interesting, but, as written, this paper will appeal to a very narrow audience of researchers who have worked or work on modeling the production/diffusion/degradation model of morphogen gradients. This stems from two reasons. First, there is a lack of conceptual explanation of what the relevant questions really are. The authors refer to a previous paper but do not explain what the arguments in that paper really were. As far as I understand, the previous paper claimed that the non-linear degradation will buffer against fluctuations in the levels of the source. That is interesting in principle, but there are other sources of noise and all of it is hard to get. I would urge the authors to rewrite their paper putting themselves in the shoes of someone who is not deeply familiar with all the details of the literature. The second reason why I find this paper confusing is that it is in part difficult to understand what the authors did. I had to check a previous paper to find some explanation of how molecular noise was introduced in the simulations. However, even after quickly checking that paper I remain confused about what the real assumptions were and how much I can trust the choice of parameters, etc. Again I would strongly recommend that the authors write a paper that is easier to read and contains all the relevant information.

---

## [Author Response]

Reviewer #1 (Recommendations for the authors):These morphogens often serve as transcription factors to help set up other, also noisy, downstream morphogen gradients. Thus, it would be interesting to have the authors' comment on whether having a linear + a non-linear or a non-linear + a non-linear set of decay functionals might change the readout accuracy further away from the source. As these calculations might be time-consuming, I'm not specifically asking for them to be done, but it would be good to get the authors' take on the matter for the Discussion section.

Unfortunately, it is unclear to us which patterning scenario the referee is referring to specifically. In a scenario where a morphogen gradient with non-linear decay establishes a second gradient with non-linear decay further away from the source, for example Hh instructing Dpp expression in the *Drosophila* wing disc [Tonimato et al. 2000, Molecular Cell], the first gradient would already lead to an imprecise readout, and thus an imprecise setup of the downstream gradient. Another possibility would be opposing gradients, as observed in the neural tube. However, we previously demonstrated that opposing gradients are not required for high patterning precision in the mouse neural tube [Vetter et al. 2022, Nat. Commun.]. Furthermore, a non-linear decay gradient could pattern the tissue close to the source and a second linear decay gradient could pattern the tissue further in the domain, but we are not aware of a biological example.

As we only see a minuscule improvement of non-linear decay in comparison to linear decay close to the source and a deterioration of precision further away from the source, we would not expect any of the mentioned combinations to be of biological relevance — a prospect that may be too limited to warrant a discussion in the paper.

My main concern with the submission is that the article is written in a very short format that is very difficult for someone outside the field (myself included) to read. All of the information was there and I was able to figure things out, but only with significant effort. I would recommend expanding the text somewhat and providing a bit more detail as to where previous mathematical approaches ended and where this one starts.

We have now expanded the manuscript to include more information about the numerical simulations and the setup. A new panel in Figure 1, for example, illustrates the French flag model, making it easier for non-experts to understand how morphogen gradients pattern a tissue. The newly added Figure 2 explains the entire modeling setup and how noise is introduced into the simulations. The previous Figure 2 has been turned into an appendix Figure 1, as it is concerned with a technical aspect that may be hard to understand by a general audience. We have also added graphical representations of the different boundary conditions to Figures 4 and 5 to make them more accessible. Moreover, we have included a free-standing methods section, further explaining how we generated the noisy gradients and discussing the choice of parameter distribution. Additionally, we have moved the majority of the mathematical derivations to a newly created appendix, which we believe makes the manuscript more accessible to a broader readership and allows the interested reader to look up the details. Finally, we explain in greater detail how the paper by Eldar et al. (2003) addresses noise and how our simulations differ from their view.

Overall, the manuscript was expanded to make it more accessible to a broader audience. Thank you for helping us improve the manuscript.

Reviewer #2 (Recommendations for the authors):1. This is a bit of author preference, but I find that I prefer a free-standing methods section as opposed to having them interleaved into the results, as they are now. I think the free-standing section allows for a bit more elaboration on details without detracting from the narrative.

We have now included a methods section to provide a more detailed explanation of the technical aspects. In the methods section, we explain how the variable gradients were generated and how the choice of parameter distribution influences the results. We separated most of the mathematical derivations from the main text and moved it to a new appendix to make the main text more accessible to a wider audience.

Thank you for pointing this out. We agree that these steps helped make the manuscript more accessible.

2. I generally found this paper a nice exploration of the question of how non-linear decay affects patterning. The paper is written in such a way that it may be challenging to read for those less familiar with the equations and conventions of mathematical modeling. If the authors wish to make the paper more accessible, I would suggest (in addition to the separation of methods) spending a bit more time defining the variables and parameters used throughout the paper, the justification for the different types of x/y axes used in the different graphs, and perhaps adding a cartoon or two illustrating a hypothetical tissue being patterned by a morphogen with some of the parameters/variables depicted. In addition, giving specific biological examples that correspond to some of the model considerations (e.g. boundary conditions) may be interesting for some readers.

We improved the manuscript in several ways. Firstly, we added a panel (Figure 1A) illustrating tissue patterning with the French flag model. Secondly, we added Figure 2, which describes the simulation setup and displays all relevant variables graphically. This makes it easier to understand how the simulations generate noisy gradients. The previous, more technical Figure 2 has been made an appendix Figure 1. Additionally, we moved the mathematical derivations to the appendix. We also added a discussion on how the choice of kinetic parameter distribution affects the results and how the variable gradients were generated to a dedicated methods section. To make the meaning of the boundary conditions clearer, we added small panels above Figures 4 and 5 that graphically illustrate them. Overall, we believe that these additions make the manuscript more informative and accessible to a broad readership also outside of the field of mathematical modeling. We thank the referee for pointing this out.

The main reason for choosing the boundary conditions is their mathematical convenience, which might also explain their widespread use elsewhere. Unfortunately, the boundary conditions in real tissues can be rather complicated as there is diffusion out of the domain, diffusion in 3D, and the boundary conditions might well differ between different systems. Here, we focused on the representation of the source in three different ways (explicit production domain, Dirichlet or von-Neumann boundary conditions as we now better illustrate (Figures 4 and 5)) while using zero-flux boundary conditions otherwise.

3. It might be interesting to add two points to the Discussion section. First, the simulations here explore the effect of parameter variation, which might be a good representation of, e.g. how genetic variability between individuals affects parameters and therefore patterning. However, even with consistent mean parameter values, molecular stochasticity can also impact these processes – is this worth exploring in future work? Second, given that most (all?) morphogens seem to operate in more complex regulatory networks, with features like opposing gradients or mutual inhibition of morphogen targets, might the interaction between non-linear decay and these other features be worth exploring in the future?

We thank the referee for pointing out these aspects, which are indeed relevant and which require further research.

We assume that by “molecular stochasticity” the reviewer refers to the effect of a low number of individual morphogen molecules on patterning precision. In this work we assume that molecule numbers remain sufficiently large to allow the morphogen concentration to be represented by a continuous field variable *C(x)*. Together with this simplification goes the assumption that cells are always able to read out a gradient concentration, even if it is very low, and that the concentration never drops to zero. Within this framework, molecular stochasticity, if interpreted in this way, is an aspect that lies outside the scope of our model.

At the moment there are only a few reliable measurements of morphogen gradients, and all that we are aware of are limited to relative (normalized) units. The absolute morphogen count remains elusive. We hope that the experimental community will soon be able to reliably measure the morphogen levels also in absolute terms (numbers of molecules). With such measurements, it would be feasible (and indeed important) to determine the influence of molecular stochasticity on gradient precision in future work. We discuss this now in the revised discussion.

In case that the referee is instead referring to stochasticity in the molecular nature of the morphogen kinetics, then this is an aspect that we hope to have made much clearer in the revised manuscript with the added illustrations and additional explanations. In our simulations, we represent such molecular stochasticity explicitly, and they are in fact at the heart of our model. Each cell in the simulated patterning domain possesses its own kinetic parameters p, d, and D, drawn independently from random distributions. It is this very aspect of our model that gives rise to noisy gradients, and to different gradients between embryos or tissues. We have revised the first section in Results, and added further details to the Methods section, to make this central aspect clearer. With the new Figure 2, we hope to also make this more visually apparent.

Regarding the referee’s second point: In this work, we focus solely on the precision of morphogen gradients and do not address their downstream readout in complex regulatory networks. Recently, it was demonstrated that opposing gradients are not required for high patterning precision in the mouse neural tube [Vetter et al. 2022, Nat. Commun.]. As we numerically show that close to the source, non-linear decay does not lead to a significant increase in precision, and that precision deteriorates further away from the source, we have no reason to assume that interactions of non-linear decay with the proposed features would significantly change the results. However, if network interactions are explicitly modeled, it would be straightforward to study the impact of non-linear decay on gradient precision using the existing framework.

We have added a new paragraph to the discussion summarizing the two important points raised by the reviewer.

Reviewer #3 (Recommendations for the authors):I find that this paper is very hard to read. The results of this paper are interesting, but, as written, this paper will appeal to a very narrow audience of researchers who have worked or work on modeling the production/diffusion/degradation model of morphogen gradients. This stems from two reasons. First, there is a lack of conceptual explanation of what the relevant questions really are. The authors refer to a previous paper but do not explain what the arguments in that paper really were. As far as I understand, the previous paper claimed that the non-linear degradation will buffer against fluctuations in the levels of the source. That is interesting in principle, but there are other sources of noise and all of it is hard to get. I would urge the authors to rewrite their paper putting themselves in the shoes of someone who is not deeply familiar with all the details of the literature. The second reason why I find this paper confusing is that it is in part difficult to understand what the authors did. I had to check a previous paper to find some explanation of how molecular noise was introduced in the simulations. However, even after quickly checking that paper I remain confused about what the real assumptions were and how much I can trust the choice of parameters, etc. Again I would strongly recommend that the authors write a paper that is easier to read and contains all the relevant information.

In the revised manuscript, we now describe in greater detail how the previous paper treated variability in the morphogen gradients as a fold-change in the morphogen influx, and how this approach is limited, as the gradients only differ by the amplitudes/fluxes, but are not variable or noisy within the tissue. Our approach allows for a broader and more realistic range of sources of noise, which indeed exists, as the referee correctly notes. Our model introduces cell-to-cell variability in all parameters (the production rate, degradation rate, diffusion coefficient, and cell areas) and also stochasticity in the morphogen influx or gradient amplitude, where applicable. We assume that these parameters follow log-normal distributions, and now provide a detailed explanation for this choice in the methods section.

We have also made several improvements to the manuscript to make it more accessible and self-contained. Firstly, we added a panel explaining conceptually how a tissue is patterned according to the French flag model (Figure 1A). In the newly added Figure 2 we illustrate the computational model with all its variables, and how noise is introduced in the simulations. Secondly, we have now added a methods section that details how the noisy morphogen gradients were generated and how the parameter distributions were chosen. Thirdly, we split the mathematical derivations off from the Results section and moved them to the newly created appendix. Fourthly, to further facilitate interpretation of the results, we now normalize the positional error by the cell diameter, which we hope is an intuitive relative scale for a broad audience. Lastly, for an easier understanding of the different boundary conditions, we added small graphical depictions of them to Figures 4 and 5. All of these changes make the paper more self-contained and accessible, hopefully allowing a much broader readership to follow the logic without consulting the references.

We thank the reviewer for helping us improve the manuscript.